# *Legionella* and the Role of Dissolved Oxygen in Its Growth and Inhibition: A Review

J. David Krause 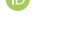

Healthcare Consulting and Contracting Inc., Tallahassee, FL 32309, USA; dkrause@hc3fl.com

**Abstract:** This review discusses the relationship between dissolved oxygen and *Legionella* growth. Growth of *Legionella* in building water systems is considered to pose a health risk and controlling it could benefit public health by reducing the number of healthcare and community acquired Legionnaires' disease cases. Conditions known to encourage *Legionella* growth include low disinfectant concentrations, warm temperature, and water stagnation. *Legionella* is an obligate aerobe meaning its cellular respiration is inhibited under anaerobic conditions. Despite evidence from earlier published studies the importance of dissolved oxygen for the survival and growth of *Legionella* has received little attention from researchers and public health authorities. Modern measurement devices can reliably determine if dissolved oxygen concentrations in potable water and other building water systems are at levels capable of supporting *Legionella* growth or inhibiting its amplification. Removing dissolved oxygen from water can be achieved using gas transfer membrane contactors. Water with low dissolved oxygen levels interferes with *Legionella*'s cellular respiration by oxygen starvation, whereas disinfectants achieve the same effect by interfering with oxygen transport across the cell membrane. Ultimately, the same effect on *Legionella* growth may be achieved by reducing dissolved oxygen as treatment with oxidants, while avoiding the drawbacks of corrosion and disinfectant byproducts.

**Keywords:** legionellosis; *Legionella* control; premise plumbing systems; treatment technologies; dissolved oxygen

## 1. Introduction

*Legionella* is a bacterium found in lakes, ponds, and streams as well as moist soils that can pose a health risk when allowed to grow or amplify in building water systems. Most species of *Legionella* can cause legionellosis in humans, but >90% of cases are attributed to *Legionella pneumophila*. Legionellosis occurs in two distinct forms, an infectious pneumonia called Legionnaires' disease, and a milder flu-like illness without pneumonia called Pontiac Fever [1,2]. Inhaling aerosolized water containing the bacteria can result in either Legionnaires' disease or Pontiac Fever [3–5] but is not transmitted from person to person.

Legionnaires' disease cases in the United States increased by over 650% over the last two decades peaking at 9,933 confirmed cases in 2018 (See Figure 1) [6]. This dramatic rise in reported cases, despite being under diagnosed and reported, was examined in a 2019 report by the National Academies of Sciences, Engineering, and Medicine (NASEM) Committee on Management of *Legionella* in Water Systems. The NASEM Committee estimated the actual number of Legionnaires' disease cases ranged from 52,000 to 70,000 cases annually in the United States, or a rate of 20.5 to 27.4 per 100,000 [7].

Dissolved oxygen (DO) is essential to the survival and growth of aerobic bacteria, including *Legionella*, in natural and human-made aquatic environments [5,8–12]. Building water systems that contain adequate levels of dissolved oxygen can support *Legionella* cellular respiration, metabolism, and growth, whereas *Legionella* and other aerobic pathogens are not typically found in anoxic groundwater deep below the water table that has no contact with the atmosphere [13]. Along with other obligate aerobic bacteria *Legionella* growth is inhibited when DO is too low to support metabolic respiration, making it a critical factor in

*Legionella*'s growth [11,12,14–16]. Reducing DO levels in building water systems could be a potential control mechanism in addition to currently recognized methods.

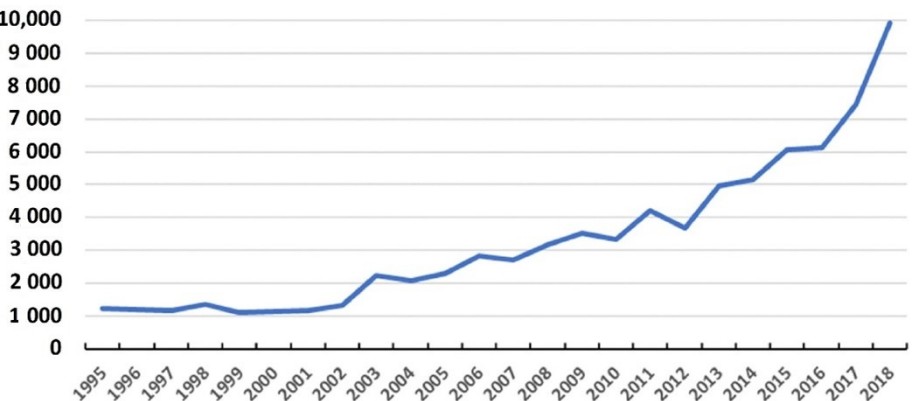

**Figure 1.** Number of Legionnaires' Disease Cases Reported Annually in the United States.

## 2. Ecology and Control of *Legionella*

*Legionella* are Gram-negative, obligate aerobic bacteria that do not exhibit secondary metabolic pathways and have been found to survive in aqueous environments with DO levels ranging from 0.2 to 15 ppm [5,10,17]. Earlier published studies have reported diminished or no *Legionella* growth under low DO or anaerobic conditions in the laboratory [11,12]. The fundamental need for adequate oxygen levels is recognized for aerobic bacteria that do not exhibit any secondary or facultative anaerobic metabolism. Feeley et al. reported that stock cultures of *Legionella* failed to grow on artificial media (agar) under anaerobic conditions but did not report the specific level of oxygen present. Wadowsky et al. reported that *Legionella pneumophila* failed to multiply under conditions of low DO, below 1.7 to 2.2 mg/L (ppm).

*Legionella* are found in many naturally occurring freshwater environments. They tend to thrive in warm, man-made water sources and are considered to be chlorine tolerant. *Legionella pneumophila*, the species associated with most cases of Legionnaires' disease, grows at temperatures between 25 °C and 50 °C (77–122 °F) with an optimal growth temperature of 35 °C (95 °F). *Legionellae* are intracellular parasites of freshwater protozoa that use a similar mechanism to infect and multiply within mammalian alveolar macrophage cells, causing respiratory disease in humans [3].

Viable *Legionella* have been found in natural reservoirs at DO concentrations ranging from 0.3 to 9.6 ppm, demonstrating its ability to survive in a wide range of aerobic environments [15]. Past research has focused primarily on temperature and disinfectant levels necessary to kill, inactivate, and control *Legionella*. Controlling and preventing the growth or amplification of *Legionella* in building water systems has historically relied upon adding and maintaining oxidizing biocides, predominately sodium hypochlorite, and other chlorine-based oxidants. Relying upon oxidant-based disinfectants in potable and utility water poses a variety of challenges. When oxidizing disinfectants bind to bacteria or react with organic debris and plumbing surfaces, they are consumed and no longer offer residual protection against bacteria. Reactions with interior plumbing surfaces increase corrosion rates that can leach lead (Pb) and copper (Cu) from wetted plumbing materials [2,18]. Adding free chlorine can increase the rate of premise plumbing water leaks and mobilize lead and copper into the water [18]. Disinfectant oxidant reactions with bacteria and other organic matter in potable water produce a number of regulated carcinogenic compounds including haloacetic acids and trihalomethanes [2].

Research on water distribution systems and premise plumbing has recently highlighted the influence of water age and water conservation measures on residual disinfectant levels and the growth of *Legionella* and other waterborne pathogens [19–21]. However,

the importance of DO on *Legionella* growth has received little attention from researchers or public health authorities, despite several lines of evidence and early published studies. *Legionella* is found to grow in tepid water with low disinfectant levels, with adequate oxygen levels and a ready supply of nutrients [5,8,12]. Water and energy conservation measures create conditions favorable for *Legionella* by increasing the residence time of water in distribution and premise plumbing lines. Increased water age consumes residual chlorine and results in lower disinfectant levels [2,20]. Low flow faucets and electronic eye (automatic) fixtures create microhabitats that harbor biofilm and *Legionella* growth [22,23].

Relying upon water temperature to inhibit *Legionella* growth in either hot or cold water systems poses many challenges. Hot water distribution systems inevitably have zones or branches where temperatures are low enough to permit *Legionella* growth. Cold water systems have no means of maintaining water temperatures below the growth range of *Legionella*, especially when ambient temperatures increase the temperature of municipal water or premise plumbing lines.

When methods currently used to control *Legionella* (i.e., effective residual chlorine levels or elevated water temperatures) fail to prevent its amplification DO levels may be a controlling factor that has been previously unrecognized.

## 3. Dissolved Oxygen in Water

DO is introduced to potable water via several mechanisms. Drinking water aeration is commonly performed to remove undesirable contaminants, such as hydrogen sulfide, radon, iron, and manganese, as part of water treatment by public water suppliers [24]. When water is exposed to the atmosphere oxygen diffusion occurs until equilibrium is reached, which is far above levels needed to support *Legionella* growth. Water treated with ozone contains higher levels of DO [14]. High-risk man-made water sources, such as hot tub spas, cooling towers, hot water systems serving large, multi-story buildings, and decorative water features use pumps to pressurize, circulate, and aerate water. Aeration processes increase the amount of DO in water and the risk of bacterial growth [24,25].

Normal DO concentrations in potable water typically range from 8 to 10 mg/L (ppm), seldom reaching 15 mg/L [26]. DO can be reduced through oxidative processes, such as the formation of iron oxide (rust) in iron pipes, or by microbial activity, when replenishment of oxygen is not possible. Natural water sources that lose contact with the atmosphere gradually become depleted of oxygen due to a combination of chemical and microbial processes. Water in deep aquifers below the water table is normally anoxic [13,27]. Anoxic conditions refer to DO concentrations of less than 0.5 mg/L (ppm), whereas hypoxic conditions are defined as being between 0.5 and 2.0 mg/L (ppm) [26,28].

*Legionella* growth is not commonly found in natural or man-made aquatic environments with low levels of DO. Lakes, rivers, and streams, with levels of DO that are limited by atmospheric pressure and temperatures tend to have low levels of detectable *Legionella*. Ground water from deep wells, typically containing little to no DO have not historically been found to be reservoirs of *Legionella* growth. Dissolved oxygen levels are typically low in water found in deep wells due to its age and lack of interaction with the atmosphere. Ground water deep below the water table is considered to be anoxic, usually containing less than 0.5 ppm [13,27]. Water in fire suppression systems, constructed of iron pipe, is deoxygenated due to rust formation which is believed to inhibit *Legionella* growth in otherwise favorable conditions [29]. When temperatures are favorable and residual chlorine levels fail to prevent *Legionella* amplification in building water systems colonization can occur. Building water systems typically associated with Legionnaires' disease, such as hot tub spas, cooling towers, decorative water features, and water heaters are also replete with dissolved oxygen.

## 4. Objectives and Relevance of This Review

The objective of this review is to examine published research on the influence of dissolved oxygen on the growth of *Legionella* bacteria and to familiarize readers with fun-

damental principles of how DO can be measured, added, and removed from natural and man-made water sources. Building water systems most frequently recognized to be at risk of *Legionella* growth include potable drinking water, potable hot water, utility water used for cooling towers, and water in hot tub spas and decorative water fountains [2–4,19]. All of these water sources typically contain substantial levels of DO. Understanding this relationship can help researchers, public health officials, industrial hygienists, engineers, and building water system operators identify systems at increased risk of *Legionella* amplification and may offer an alternative method of controlling its growth [2,30,31].

Many underlying factors and technical reasons for the observed increase in Legionnaires' disease rates have been hypothesized, but public health agencies and researchers have not reached consensus. Public health officials and researchers have examined the possible influence of increased surveillance, case ascertainment, and the use of urine antigen testing to diagnose patients on the increased number of reported cases of Legionnaires' disease. While the authors of a 2007 article concluded that increased rainfall was likely the cause, a consensus on the matter has not been reached [32].

Until a definitive reason for the increasing incidence of Legionnaires' disease is determined building operators are being urged to implement water management programs to monitor, control, and mitigate building water systems to prevent disease outbreaks caused by waterborne pathogens such as *Legionella* [33,34]. Inadequate residual disinfectant levels are a commonly identified factor that increases the risk of *Legionella* growth and has been linked to many outbreaks [31,35]. Many factors can deplete disinfectant residual in premise plumbing and other building water systems. Implementing efforts to maintain adequate residual disinfectant levels that can control bacteria throughout building water systems is currently a focus of many public health guidance documents [2,33,34]. Despite publication of numerous guidance documents and standards supporting the implementation of water management programs in 2015 to control *Legionella* in building water systems the number of reported Legionnaires' disease cases and outbreaks have continued to rise, from 6079 in 2015 to 9933 in 2018, constituting a 63% increase over 3 years [6].

The 2015 publication of ASHRAE Standard 188 and the CDC Toolkit on *Legionella*, represent a fundamental shift for public health authorities recommending a preventive approach. The June 2017 directive from the Centers for Medicare and Medicaid Services (CMS) mandated all hospitals and skilled nursing facilities who receive Medicare funding to implement water management programs that consider the ASHRAE Standard and CDC guidance [34]. ASHRAE standard 188 describes an approach to disease prevention that relies upon monitoring for and controlling conditions that encourage the colonization and growth of *Legionella* [33]. Increased monitoring for waterborne pathogens will likely reveal sources of *Legionella* growth in building water systems and prompt mitigation and remediation measures. Controlling *Legionella* growth has primarily relied upon increasing hot water temperatures and treating potable water systems with supplemental disinfectants. The practical efficacy of increasing hot water temperatures is limited due to scalding risks if delivery temperatures exceed 130 °F [2,31]. Adding oxidizing disinfectants to premise plumbing water systems can corrode plumbing materials and equipment, potentially leaches lead (Pb) and copper (Cu) into the potable water and generates disinfectant by-products (DBPs) that pose chronic health risks [2,36]. Adding supplemental chemical disinfectants to potable drinking water systems in buildings often triggers regulatory oversight for most facilities under the Safe Drinking Water Act (40 CFR Part 141 Subpart L 141.130 of the US EPA SDWA).

Ultraviolet (UV) irradiance of drinking water is also a recognized treatment technology capable of inactivating *Legionella* in the water that flows through a UV reactor. Because UV irradiation only effects bacteria in water that flows past the lamp, there is no residual effect on bacteria or biofilm established downstream of the point of treatment. Bacteria encased within debris or amoeba can be shielded from the UV irradiance, allowing for some microorganisms to escape treatment when "shaded" or otherwise protected from the UV energy. Where *Legionella* has already colonized premise plumbing lines and water

heaters UV treatment would have no effect. When UV disinfection is used to treat drinking water that has residual chlorine disinfectant it can diminish the residual chlorine levels, potentially increasing the risk of *Legionella* amplification downstream of the treatment. For these reasons UV treatment is generally not used as a standalone method for building water systems but is often applied at the point of use or to treat water after carbon filtration in conjunction with other treatment methods. Supplemental injection of chlorine-based chemicals (i.e., sodium hypochlorite, chlorine dioxide and monochloramine) have dominated the technologies used to treat building water [2]. While UV disinfection can be used as a supplement to other treatment technologies, its application is typically site and system specific.

Recognizing DO as an important factor to the growth and survival of *Legionella* could improve our ability to assess a water system's risk of contamination. Adding DO measurements to *Legionella* source assessments and prevention efforts can aid industrial hygienists, infection control professionals, and environmental health professionals to characterize a water source's amplification risk. Along with measuring water temperature, disinfectant concentrations, and pH, the concentration of DO can help to identify if conditions in a water source are favorable or unfavorable for *Legionella* growth.

## 5. Discussion of Published Studies

Published studies that reported DO levels in water sources or laboratory media and examined *Legionella* growth were sought. In the period following the discovery of *Legionella* as the cause of human disease, research primarily focused on variables that encouraged its growth in natural habitats, laboratory water sources, and on culture media. Only a handful of studies examined the association of DO in water and the growth or amplification of *Legionella*. However, it has been long recognized that because *Legionella* is an obligate aerobe it requires dissolved oxygen for survival and growth [4,5,17,37].

The first published article to report *Legionella* growth inhibition under anaerobic conditions was by Feeley et al. in 1978. The authors reported that stock cultures of *Legionella* growth on artificial media (agar) was decreased with each reduction in $O_2$ and failed to grow under anaerobic conditions [11]. At the time, researchers were most interested in defining the conditions that best supported *Legionella* growth in the laboratory and did not report any further exploration of its dependence on DO or its possible role in controlling growth in building water systems.

Two published studies by Fliermans et al. examined the ecological distribution of *Legionella pneumophila* in natural water bodies along with measuring DO levels [15,16]. The authors examined over 200 samples from twenty-three lakes in the southeastern United States as possible natural habitats for *Legionella pneumophila*. Their research simply reported water parameters found in each lake sample that included temperature, conductivity, pH, DO, and *Legionella pneumophila* concentrations. Four of the lakes were confirmed to contain *L. pneumophila*, with DO levels ranging from 3.80 to 9.80 ppm. DO concentrations were not reported for samples that did not contain *Legionella* [16].

The research was expanded to examine 793 samples from sixty-seven lakes and rivers in the southeastern United States. This study reported water parameters for each water sample that included temperature, conductivity, pH, DO, chlorophyll a, pheophytin, water clarity, and *Legionella pneumophila* concentrations. The importance of DO was not examined in detail, but the authors concluded that a relationship between *L. pneumophila* and algal photosynthesis existed. Of the 47 samples positive for *L. pneumophila* DO ranged from 0.3 to 9.6 ppm, with a geometric mean of 4.3 ppm, and 90% of positive samples having DO levels above 1 ppm. Only 10% of *L. pneumophila* positive samples had DO levels between 0.3 and 1 ppm. DO levels in samples that were undetectable for *L. pneumophila* were not reported. The warm habitats with algal products may constitute a natural habitat for *L. pneumophila* [15]. Because oxygen is a major product of algal photosynthesis, it would be expected to create an aerobic aquatic environment.

While evaluating a method to maintain water cultures of naturally occurring *Legionella pneumophila*, Wadowsky developed a model system to determine the effects of temperature, pH, and DO on its multiplication in potable water. The 1985 study focused on how to successfully grow *Legionella* but was the second to describe a correlation between low DO concentrations and *Legionella* growth inhibition. The researchers reported that *L. pneumophila* was able to multiply in their system at saturated levels of DO, with viable concentrations increasing by 1.0 Log in 7 days. However, they found that *L. pneumophila* failed to multiply under anaerobic conditions, with viable concentrations decreasing 1.7 log after 28 days. Reported DO concentrations ranged from 1.7 to 2.2 mg/L (ppm) under "anaerobic" conditions. The authors proposed that naturally occurring *Legionella pneumophila* may survive under prolonged anaerobic conditions, but that amplification does not occur [12].

A study of cooling towers in Puerto Rico reported data that included DO levels and concentrations of *Legionella pneumophila* serogroup 1 (LP1) [38]. This study also reported water parameter measurements including temperature, conductivity, pH, alkalinity and hardness in addition to *Legionella pneumophila* concentrations. While not the focus of this study the data reported on DO concentrations in all five cooling towers were in a range that supports *Legionella* growth and aerobic respiration, from 4.0 ppm to 8.8 ppm. Concentrations of *L. pneumophila* ranged from 6,300 to 23,000 cells/mL, which coincidentally correlated with DO levels ($R^2 = 0.98$) (See Figure 2). However, such a small data set and with all DO levels in the range that supports *Legionella* growth, any conclusions about the role of DO at lower concentrations cannot be supported by this study.

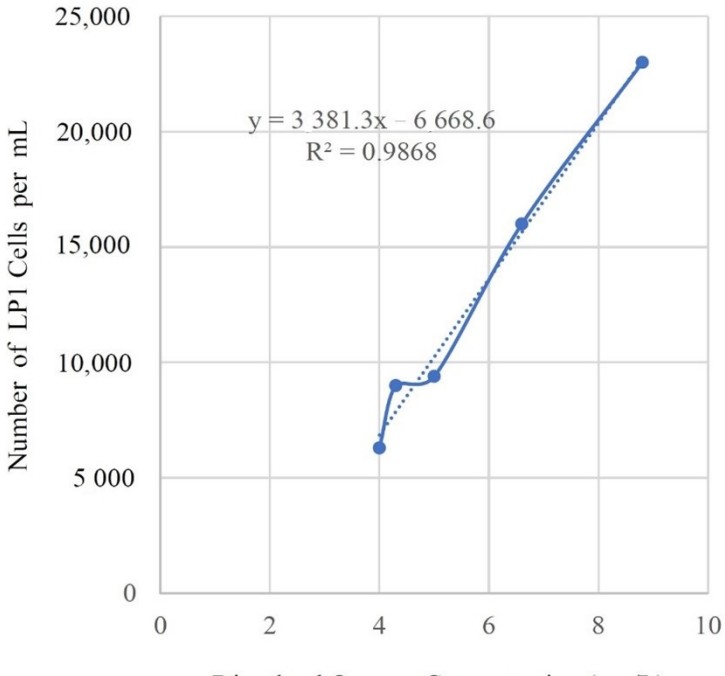

**Figure 2.** LP1 vs. Dissolved Oxygen in Cooling Tower Water.

There were likely many factors influencing the reported concentrations of *L. pneumophila* in each cooling tower. The small number of samples, the analysis methods used to detect *Legionella*, and methods to measure DO suggest a study of low power. Despite the many technical and statistical limitations of this small data set, the relationship between DO levels and reported concentrations of *Legionella pneumophila* warrants further research.

In 2001 the US EPA updated the 1999 and 1985 Human Health Criteria Document on *Legionella* citing a study by Nguyen et al. regarding the ability of *Legionella* to survive in

water with DO levels above 0.2 ppm (1991) and work performed by Yee and Wadowsky published in 1982. The latter study suggested that oxygen availability may have been growth-limiting in their experiment. The authors speculated that *Legionella* may be growing in cooling towers and other water sources where a large air-water surface interface exists, and oxygen is able to diffuse into the water. Yee and Wadowsky speculated that limited diffusion allowed through metal tubes used in their experiment inhibited *Legionella* growth [39]. Unfortunately, the DO concentrations present in the water were not reported in this study.

A 2006 study by Liu et al. examined the effect of stagnation on *Legionella* colonization that used a model plumbing system with various flow rates and turbulence. The authors found that stagnant conditions did not actually promote *Legionella* colonization, with the lowest counts of *Legionella* found in water from pipes with stagnant flow. While the study did not include measurement of DO, the authors proposed that turbulent conditions increased oxygen and nutrient availability at the attachment surface where biofilm develops, and in stagnant lines the lack of oxygen and nutrient limited its growth [40].

Findings of a study to improve a hospital's hot water system hydraulics through temperature monitoring to reduce hazards associated with *Legionella* included measurements of DO [41]. The authors concluded that temperature and turbidity were dominant predictive variables for the building water systems evaluated, but pH and DO were not predictive variables. However, the DO levels reported in the water systems ranged from 6.1 to 10.0 ppm, with median values of 8.1 and 7.1 ppm. All of the DO measurements reported in this study were far above levels reported by others to inhibit *Legionella* growth. Despite the authors' finding that DO was not a predictive indicator, the range of DO levels reported in this study were above the biological threshold suggested in earlier published studies by Wadowsky and Fliermans.

In 2016 the US EPA published a scientific literature review on Technologies for *Legionella* Control in Premise Plumbing Systems. Low DO levels were described as one of several factors that could induce a viable but not culturable (VBNC) state in *Legionella*. Chemical and environmental factors that were reported to induce VBNC in bacteria included nutrient starvation, temperature, high salt concentration, low oxygen concentration, heavy metals, pipe material, and treatment with chemical disinfectants [2,42]. While the EPA review did cite the study by Wadoswky et al. (1985) that found *L. pneumophila* amplified at dissolved oxygen concentrations ranging from 6.0 to 6.7 mg/L (ppm), and that *Legionella* growth did not occur when DO was less than 2.2 mg/L (ppm), there was no discussion of lowering DO as a potential application to control *Legionella*.

Brigmon et al. published results of analyses comparing *Legionella* concentrations in cooling towers at the US Department of Energy's Savannah River in Aiken, South Carolina. The research explored environmental factors and conditions that may have contributed to increased *Legionella pneumophila* levels, and variability in control measures over time. A model was developed using thirteen months of water chemistry and microbial data in five cooling towers. The study found that one tower had statistically higher *Legionella pneumophila* levels and that lower levels of DO and free chlorine were the most likely cause. Increased levels of accumulated debris in the cooling tower basins and higher than normal rainfall were offered as contributing factors [43].

The model they developed suggested that as DO levels increased, the *Legionella* concentrations decreased. However, all DO concentrations measured in the cooling towers were reported to be in the range considered to be well oxygenated and able to support *Legionella* growth. The reported range of dissolved oxygen evaluated by the model was from ~8.5% to 100% saturation, equating to 0.5 mg/L to 7.7 mg/L at 30 °C. The authors concluded that as DO and/or Free Chlorine levels increase the levels of *Legionella* in a cooling tower decreases. Conversely, the model predicts that as DO and/or Free Chlorine levels decrease, the level of *Legionella* would increase. However, the authors did not integrate levels of DO below 0.3 mg/L or discuss the inverse relationship between DO and free chlorine levels in cooling towers. The mechanism of dissolved oxygen introduction for cooling towers is

primarily aeration, which occurs as water drains across the fill media and when make-up water is added from municipal water supplies. As evaporative cooling occurs both heat and free chlorine are lost, and oxygen diffuses into the water. The inverse relationship between free chlorine loss and the increase in DO is not explained, considered, or discussed. The authors do not disentangle the two parameters and fail to consider that increased DO is an artifact of cooling tower operation and inherently occurs as free chlorine is lost to the evaporative cooling process. While oxidative stress may account for *Legionella* inhibition at artificially high levels of DO, the interaction of free chlorine's biocidal effects with available oxygen is a well-recognized process known to inhibit *Legionella* growth and respiration.

## 6. *Legionella* Growth and Human Disease

*Legionella* growth in building water systems is a key link in the chain of events leading to Legionnaires' disease. The mere presence of detectable *Legionella* has not been found to be a reliable indicator of increased disease risk [30,31]. According to the Centers for Disease Control and Prevention (CDC), the amounts of *Legionella* found in freshwater environments, such as lakes and streams, have not been found to cause disease. In order for *Legionella* to pose a health risk, it must first grow (increase in numbers), then become aerosolized in contaminated water droplets [44]. Environmental factors that inhibit *Legionella* growth in building water systems, and can be controlled, have the potential to reduce sources of exposure and the risk of contracting Legionnaires' disease.

## 7. Dissolved Oxygen in Building Water Systems

DO is the molecular form of oxygen present in water that is available to microorganisms and higher organisms for respiration. The oxygen atoms in water ($H_2O$) do not contribute to DO levels because they are tied up in a molecular compound. DO is critical to fish, invertebrates, bacteria, and fungi. Areas where DO levels in lakes and oceans are too low to support aquatic organisms are referred to as "dead zones" [45].

Oxygen has limited solubility in water, as described by Henry's law. The amount of oxygen that can be dissolved in water is directly related to atmospheric pressure and inversely related to water temperature and salinity. This means that cold water can hold more DO than warm water. Additionally, water under increased pressure can hold more DO than water at atmospheric pressure. At sea level, the solubility of oxygen ranges between 7.0 and 10.0 mg/L at temperatures ranging between 15 to 31 °C [13].

DO is measured in water using either an electrochemical or optical sensor. An early chemical analysis method called the Winkler titration provided a good estimate of DO but was subject to human error and is difficult to perform in the field. Both electrochemical and optical sensors can automatically adjust for temperature, pressure, and salinity as they can affect the concentration of DO in water [46,47]. Electrochemical sensors, also referred to as amperometric or Clark-type sensors, are reported to have accuracies ranging from 0.01 to 0.1 mg/L and response times ranging from seconds to a few minutes. With intelligent electrochemical or optical DO sensors capable of automatically adjusting for temperature, pressure, and salinity real-time monitoring of DO is available [48]. Due to technical advances in optical sensors their use has been widely adopted and have supplanted electrochemical sensors for field measurements [47].

Hot water in a recirculating premise plumbing system experiences conditions not typically found in nature. Heated water is pressurized and circulated throughout the sealed pipes of a building. Water under pressure is able to hold more DO, but heating it allows the dissolved oxygen to rapidly come out of solution once released from the pressurized lines. Expansion tanks and other voids or dead-leg pipes can allow air bubbles to coalesce in distal pipes where temperatures drop. *Legionella* growth may be occurring at these air-water interfaces, both in distal fixtures and within the circulating premise plumbing lines. Additional DO is regularly introduced to a circulating hot water system via the cold make-up water that replaces water discharged from faucets and sinks. This cyclic process can result in hot water that contains higher levels of DO than it could under atmospheric

conditions. The results of this supersaturated water can be observed when hot water is discharged into a clear glass and the visible haziness initially present forms effervescent bubbles that rise as the dissolved gasses, including oxygen, comes out of solution at the lower atmospheric pressure. In these man-made water systems, the amount of DO available to bacteria can be greater than in natural water sources.

Another mechanism that can lead to higher levels of DO in drinking water is ozone treatment. Municipal water systems are sometimes treated with ozone to address a wide variety of inorganic, organic and microbiological problems as well as taste and odor problems. Ozone has been found to be effective against bacteria, viruses, and protozoans, including *Giardia* and *Cryptosporidium*. Despite offering no residual effectiveness, ozone treatment of municipal water supplies is commonly used. An unintended consequence of ozone treatment is that water can become supersaturated with DO, reaching levels of 15 to 50 mg/L [2,49].

DO in water can be removed by chemical, biological, or physical means [47]. Stagnant water lines made of iron, such as fire suppression systems, become anaerobic over time due to iron oxide formation [29]. In potable water distribution systems that use monochloramine as a secondary disinfectant, nitrification can occur. Nitrification is a microbial process where ammonia is oxidized to nitrite and nitrate. The process of nitrification consumes dissolved oxygen and can deplete DO levels inhibiting nitrogen reducing bacteria [50]. To remove DO in boiler water so that corrosion of metal is prevented was historically achieved by adding a rapid oxidizer such as hydrazine [51]. In water sources without atmospheric contact the aerobic bacteria that are present consume dissolved oxygen for metabolic respiration. This irreversible biochemical redox reaction reduces DO levels to such a low concentration that continued bacterial respiration ceases [13]. Most of the chemical processes that remove DO from water can leave it unpalatable or toxic, with the exception of deep well water.

In the past twenty years membrane contactors using gas transfer membranes have been developed to degas fluids for sensitive manufacturing processes and to prevent corrosion in boilers and radiant heating systems [52]. A membrane contactor system removes DO and other dissolved gasses through physical mechanisms of vacuum and rapid diffusion that do not adversely impact water potability. Commercially produced membrane contactors are used for bottled beverage production (i.e., beer, wine, juice) and pharmaceutical manufacturing to remove oxygen. Using a membrane contactor DO levels can be efficiently reduced in water from a level of 8 parts per million (ppm) to 1 part per billion (ppb) or less [53]. Thermal treatment to remove DO from water requires tremendous energy and specialized equipment to treat batches of water. Membrane contactors represent the only practical technology that is currently available to treat drinking water and other building water systems without rendering them undrinkable or hazardous [53,54].

Careful consideration and research should be made of the potential shift in microbial community and the water chemical parameters of premise plumbing systems operated under reduced dissolved oxygen conditions. The interaction between existing plumbing materials and equipment as well as chemical disinfectants introduced from municipally treated water need to be examined in any pilot or test system. Examining changes in the types of waterborne pathogens present in premise plumbing systems is needed to detect if DO reduction unintentionally creates a favorable habitat for other microorganisms while inhibiting *Legionella* and other aerobic pathogens.

## 8. Bacterial Control by Inhibiting Cellular Respiration

Two published articles have reported that oxygen starvation inhibits *Legionella* growth in laboratory studies [11,12]. While cell death may not rapidly occur, inhibiting cellular respiration does prevent *Legionella* amplification and can eventually cause cellular respiration to permanently cease. The mechanism believed to inhibit bacterial growth is oxygen starvation due to the lack of dissolved oxygen in the environment which prevents cellular respiration, ceasing the production of adenosine triphosphate (ATP) within the cell. Similarly, *Legionella* growth inhibition by commonly used disinfectants, including hypochlorous

acid (aka free chlorine), ozone, and copper-silver ions, is also believed to be the result of interfering with cellular respiration, presumably by disrupting oxygen transport across the cell membrane [55–58].

Free chlorine's effects on *Legionella* are believed to be dose dependent, causing both lethal and non-lethal damage depending upon concentration and duration of exposure. Scientific consensus has not been reached on the exact mechanism that free chlorine inactivates or kills bacteria, including *Legionella*, at concentrations typically found in drinking water [57,59]. Adding sodium hypochlorite to form the active compound hypochlorous acid is often used as a biocidal control strategy for *Legionella* and other waterborne pathogens. *Legionella* may remain viable but nonculturable at hypochlorite concentrations up to 100 ppm [60,61].

Studies suggest that active transport and respiration systems for glucose and amino acids are targets of chlorine-induced cell injury. Chlorine-induced injury to cell membranes reduces membrane potential and respiratory activity, significantly reducing bacterial culturability [56]. Chlorine can inactivate bacteria through a number of pathways, including sulfhydryl enzyme oxidation, amino acid ring chlorination, loss of intracellular contents, decreased uptake of nutrients, protein synthesis inhibition, respiratory component oxidation, decreased ATP production, DNA damage, depressed DNA synthesis, or decreased oxygen uptake. The actual mechanism by which chlorine inactivates bacteria could involve a combination of these factors [62]. Regardless of the precise mechanism that chlorine and other oxidizing disinfectants rely upon, a common method to determine if bacteria have been inactivated or killed is to measure cellular respiration byproducts [55,63]. Similarly, the mechanism of action that copper and silver ions use to inactivate *Legionella* is believed to be interference with enzymes involved in cellular respiration [64].

## 9. Potential Changes to Microbiome

The long-term consequences of reducing dissolved oxygen in building water systems should be considered. Many of the microorganisms that currently cause waterborne illness are aerobic (*Campylobacter*, *Legionella*, *Nontuberculous mycobacteria* (NTM)). Waterborne pathogens that are facultative anaerobes (*Cryptosporidium*, *Norovirus*, *Giardia*, *Pseudomonas aeruginosa*, *Streptococcus pneumoniae*) result from fecal contamination [65] and would continue to be of concern if such contamination occurs. A relevant example of the types of bacteria that can become established under anaerobic conditions are those that colonize iron pipes of fire sprinkler systems. The bacteria of concern reported in fire sprinkler lines include iron reducing bacteria (IRB) and sulfate reducing bacteria (SRB), both of which grow under anaerobic conditions where the sole electron acceptor is either Fe(III) or Mn(IV). However, the presence of sulfate or its microbial byproduct, hydrogen sulfide, would be readily recognized by the disagreeable odor and iron pipes are no longer used for premise plumbing drinking water. Further research on the types of microbial communities that may form within building water systems and premise plumbing lines under hypoxic or anaerobic conditions would be needed.

## 10. Future Directions: Controlling Dissolved Oxygen to Control *Legionella* Growth

Research on using reduced DO as a control measure for building water systems could potentially offer a novel method to prevent *Legionella* colonization and growth. The time it takes for *Legionella* inhibition to occur at DO levels below 0.2 ppm, and whether a threshold above or below 0.2 ppm DO can achieve the same effect, have not been thoroughly investigated [17]. It is recognized that DO is transported across the cell membrane of aerobic bacteria and used for metabolic respiration [5,9,13]. When sufficient levels of DO are unavailable, or other conditions inhibit metabolic respiration, *Legionella* can enter a dormant or viable but not culturable (VBNC) state. During the transition from the culturable state to the VBNC state, *Legionella* bacterial cells reduce nutrient transport and respiration activity [66]. *Legionella* in the VBNC state is unable to cause infectious disease [67,68]. Recovery from the VBNC state can occur, but in vitro and animal studies suggest that the organism's virulence

is substantially reduced if it does recover, and that direct infection of human cells has never been observed from VBNC *Legionella* [42,69,70]. Complete loss of viability, through any mechanism, accompanies an inability to cause infection [71].

*Legionellae* that have undergone phagocytosis by amoebae are believed to be protected from disinfectants in drinking water while encased within the amoebae. *Legionellae* are able to withstand and survive chlorine treatment when contained within amoeba [2]. It has been reported that *Legionella* can replicate within phagolysosomes and become more virulent [72]. The presence of amoeba in water sources that also harbor *Legionella* makes it more difficult to treat, control, and prevent *Legionella* from re-colonizing. While it is recognized that amoebae generate reactive oxygen species within phagolysosomes the dissolved oxygen levels have not been reported.

Experimental data demonstrating the effects of anaerobic conditions or low DO levels on *Legionella* in a model or actual premise plumbing system is needed to determine if it is either feasible or practical as a control method. Earlier reports of the relationship between lower DO levels and *Legionella* inhibition were limited by their observational nature and a focus on finding conditions that encouraged *Legionella* growth on media or in aquatic culture rather than those that inhibited *Legionella*. The threshold levels of DO leading to transient inhibition, bacterial dormancy, transition to a VBNC state, permanent inactivation, or cell death have not been reported for *Legionella* under anaerobic conditions. The duration of exposure to water with low DO levels that is necessary to effectively inhibit *Legionella* growth within a premise plumbing system or to effectively reduce established colonies of *Legionella* needs to be experimentally determined. The effects of lowered dissolved oxygen levels in premise plumbing systems needs to be studied to determine if shifting it to a hypoxic or anaerobic condition affects the levels of metals in water, dissolved organic carbon, or other physical chemistry impacting water quality. It is reasonable to anticipate that reduced corrosion and oxidation of plumbing materials would be observed, but other unanticipated consequences would need to be examined. If found to safely and effectively inhibit *Legionella* amplification these data could then be applied to systems, technologies, and practices for controlling *Legionella* in building water systems moving forward.

Inhibiting *Legionella* growth with traditional disinfectants is analogous to that of oxygen starvation, with both approaches essentially interfering with *Legionella*'s ability to carry out cellular respiration. Whether cellular respiration is inhibited by interfering with oxygen transport across the cell membrane or because oxygen is unavailable in the aqueous environment, the same endpoint of *Legionella* inhibition could be achieved. When levels of DO are above the minimum required threshold, *Legionella* is able to carry out metabolic respiration and amplification, so long as other environmental factors are favorable. Limited evidence suggests that under enhanced DO levels, *Legionella* may become more virulent and pose an increased rate of amplification. Measuring DO concentrations when examining possible amplification sources during outbreak investigations could shed more light on the relationship between DO and the growth of *Legionella* or its inhibition in certain building water sources.

## 11. Author's Perspective

Current methods and approaches to control *Legionella* in building water systems have generally followed those used to treat municipal drinking water and the distribution systems. Constantly adding oxidizing disinfectant biocides carries with it many drawbacks described earlier and does not fit the way that water is used in building water systems and premise plumbing. Of great concern is that supplementing building drinking water with chlorine-based oxidants usually requires water to be flushed from distal fixtures and runs contrary to water conservation needs. Exploring alternative approaches to control *Legionella* growth in building water systems, such as reducing dissolved oxygen levels and other mechanisms could overcome many of the adverse consequences of supplemental disinfectant treatment that include creating carcinogenic byproducts, increasing premise plumbing corrosion, and wasting water.

## 12. Summary

DO is critical to the growth and survival of *Legionella* in aqueous environments and building water systems. DO can be reliably measured using direct reading instruments and probes to determine if concentrations are low enough to inhibit or above levels that could promote *Legionella* growth. Based upon most of the available literature DO levels below 0.3 ppm can prevent *Legionella* growth in water by inhibiting cellular respiration. While cell death or complete loss of viability may take some time, growth of obligate aerobic bacteria can be controlled or prevented under anaerobic conditions. *Legionella* growth in building water systems poses a health risk and is a key link in the development of Legionnaires' disease. Published research has demonstrated that in natural water sources and laboratory experiments low DO concentrations inhibited *Legionella* growth. Preventing or inhibiting *Legionella* growth in building water systems is believed to lower the risk of Legionnaires' disease in humans. Measuring DO in suspected water sources could be integrated into future *Legionella* risk assessments. Continuous real-time monitoring of DO in building water systems could provide an early warning of increased *Legionella* growth risk. Reducing or eliminating DO in building water systems could offer a promising alternative to inhibiting *Legionella* growth in premise plumbing, hot water systems, and utility water sources. Effectively controlling *Legionella* amplification in water sources by removing DO, could reduce the risk of Legionnaires' disease while potentially avoiding corrosion and disinfectant byproducts associated with oxidant chemical treatment methods. Membrane contactors to remove DO from liquids represents the only currently available technology that can practically reduce dissolved oxygen in drinking water because they do not adversely impact water potability. Alternative methods to reduce dissolved oxygen in water rely upon adding chemicals that are toxic or would adversely impact the water taste or require boiling the water under vacuum. Experimental data demonstrating *Legionella* control in building water systems by reducing DO are needed to determine if this could be used as an effective and practical control method.

## 13. Patents

Patent Number US 10,913,663 B2 was awarded on 9 February 2021 to the author on Systems and Methods for Controlling Waterborne Pathogens.

**Funding:** Preparation of this review received no external funding.

**Institutional Review Board Statement:** Not Applicable.

**Informed Consent Statement:** Not Applicable.

**Data Availability Statement:** Not applicable.

**Conflicts of Interest:** The author declares no conflict of interest.

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
