# Peer review of "Legionella and the Role of Dissolved Oxygen in Its Growth and Inhibition: A Review"

_water, doi:10.3390/w14172644_

Round 1

Reviewer 1 Report

General comments

Abstract: Suggest elaborating on concluding remarks; 1) practicality of DO removal in a premise plumbing application and 2) benefits to public health.

Section 1: Besides Legionella and other aerobic pathogens, suggest elaborating on anaerobic bacterial pathogens and impact of microbial community shift due to lowering DO which is associated with microbial risks.

Section 2: More rigorous discussion about Legionella occurrence in building water systems (Line 78-80) is needed. According to previous published studies, Legionella are often detected in hot water of premise plumbing water systems, but not often in chlorinated cold water, possibly due to low chlorine residuals in hot water. In this case, DO does not seem to play a critical role in Legionella control.

Section 4: As stated in the text (Line 191-197), oxidizing disinfectants including chlorine can cause corrosion of pipes, potentially leaching lead and copper into potable water. Indeed, environmental condition changes (e.g., physicochemical changes) could shift water quality parameters including microbial community/population and chemical composition like what happened in Flint, MI. Strongly suggest elaborating on any environmental impact due to anoxic or anaerobic condition after DO reduction.

Section 5: This is the main section which discussed previous published studies. However, as stated in the main text, most studies cited here focused on ecology of Legionella in natural habitats (Line 238-257) and Legionella growth on culture media (Line 231-237). In the rest of this section, the 2016 USEPA review report (Line 315-324) and Legionella in cooling towers (Line 325-353) were summarized. The EPA report did not include lowering DO to control Legionella in the candidate tool list. Rather, the report discussed Legionella growth at different levels of DO. Any cooling tower studies did not discuss DO and Legionella control, but chlorination as a primary parameter to control Legionella. Strongly suggest adding more literature about association of DO for Legionella control in premise plumbing systems (PPS).

Section 7: This section mainly discussed two points: 1) (Line 387-401) In hot water in a recirculating PPS, DO levels are higher than natural water sources and 2) (Line 423-434) Membrane technology to remove DO and its application in beverage production and pharmaceutical manufacturing. However, careful consideration regarding “microbial community shift” and “characterization of chemical water quality” under changed environmental conditions after drastic DO removal must be given to its application in PPS.

Section 9: Besides summary of Legionella VBNC state (Line 474-483), suggest providing additional discussion about Legionella growth in phagocytic amoeba which could provide low DO environment and additional protection in chlorinated water?

Specific comments

Line 145-154: Redundant. This paragraph is almost identical to Line 34-39.

Line 479: ‘invitro’ should read “in vitro” and italic as well.

Reviewer 2 Report

Krause reported review article "Legionella and the Role of dissolved oxygen in its growth and inhibition: A Review" is well written and can be published in Water upon addressing minor comments as specified below. 

1) Some of the recent references related to Biological oxygen demand are missing in the introduction. 

2) Better to provide biochemical pathways through the pictorial representation before summary. 

3) This article is comprehensive review so it is desirable to provide author's perspective in the last section.  

Round 2

Reviewer 1 Report

I have no further comments.